# Effect of Different Feed Particle Size on Gastrointestinal Tract Morphology, Ileal Digesta Viscosity, and Blood Biochemical Parameters as Markers of Health Status in Broiler Chickens

**DOI:** 10.3390/ani13152532

**Published:** 2023-08-05

**Authors:** Jakub Novotný, Lucie Horáková, Michal Řiháček, Dana Zálešáková, Ondřej Šťastník, Eva Mrkvicová, Vojtěch Kumbár, Leoš Pavlata

**Affiliations:** 1Department of Animal Nutrition and Forage Production, Faculty of AgriSciences, Mendel University in Brno, Zemědělská 1, 613 00 Brno, Czech Republic; jakub.novotny@mendelu.cz (J.N.); xhorak34@mendelu.cz (L.H.); xrihace2@mendelu.cz (M.Ř.); xzalesa4@mendelu.cz (D.Z.); ondrej.stastnik@mendelu.cz (O.Š.); eva.mrkvicova@mendelu.cz (E.M.); 2Department of Technology and Automobile Transport, Faculty of AgriSciences, Mendel University in Brno, Zemědělská 1, 613 00 Brno, Czech Republic; vojtech.kumbar@mendelu.cz

**Keywords:** intestine morphometry, villus height, crypt depth, poultry nutrition, feed mixture physical characteristics

## Abstract

**Simple Summary:**

The article deals with the impact of different feed particle sizes in diets on the morphology and morphometry of the gastrointestinal tract and the viscosity of digesta as an indicator of the health status of the gastrointestinal tract and the whole organism. The health of the intestine is given considerable attention in human nutrition due to various autoimmune and other diseases. Our goal was to focus attention on this issue in animals as well. This study looks at how physical structure affects the health of broilers. This article also evaluates selected blood biochemical parameters. The performance, consumption, and conversion of feed were monitored as additional parameters, i.e., basic monitored parameters in feeding trials. The use of coarse feed particle size in the diet had a positive effect on the gizzard weight and small intestinal villi height and crypt depth, which increased the surface area intended for digesting nutrients. The use of finely ground particles in the feed increased the level of gamma-glutamyl transferase and at the same time, decreased the level of urea, which could indicate adverse changes in the liver. The performance parameters, feed intake, and feed conversion ratio were not affected by using different feed particle sizes.

**Abstract:**

The study is focused on how the physical structure of the feed affects the health status of broiler chickens. The aim of this study was to evaluate the influence of feed particle size in broiler diets on gastrointestinal tract morphology, digesta viscosity, and blood biochemical parameters. A total of 90 one-day-old male Ross 308 broiler chickens were randomly divided into three different experimental groups (with five replicates per pen), with 6 birds per cage. The first experimental group (Coarse) was fed with the coarsest particle size, with feed with a geometric mean diameter (GMD) of 1111.26 µm, the next group (Medium) was fed with a less coarse feed size of GMD 959.89 µm, and the last group (Fine) was fed a diet with a fine feed particle size of GMD 730.48 µm. The use of coarse feed particle size in the diet had a positive effect on the gizzard weight and small intestinal villi height and crypt depth, which increased the surface area intended for digesting nutrients. The use of finely ground particles in the feed increased the level of gamma-glutamyl transferase and at the same time, decreased the level of urea, which could indicate adverse changes in the liver.

## 1. Introduction

The health of the gastrointestinal tract is one of the important indicators of the overall health of an organism. The health of the intestine is given considerable attention in human nutrition due to the existence of various autoimmune and other diseases. Intestinal epithelial barrier dysfunction and increased permeability result in a “leaky gut”, which is associated with intestinal disorders in humans, such as inflammatory bowel disease (IBD), bowel syndrome, irritable liver disease (ILD), fatty liver, steatohepatitis, liver cirrhosis, and collagen diseases. An imbalance in the qualitative and quantitative composition of the intestinal microflora, or dysbiosis, contributes to the dysfunction of the intestinal barrier and leaky gut syndrome [1]. Intestinal hyperpermeability resulting from leaky gut syndrome causes alterations in tight junctions, allowing toxic substances to enter the bloodstream, leading to dysfunction in various organs and systems. Stimulation of the immune system [2,3,4], degradation of fiber, and an increase in the function and motility of the gastrointestinal tract facilitate the absorption of nutrients and pathogen inhibition. The intestine is involved in maintaining immunity and repairing the gastrointestinal mucosa [2]. In this study, we attempt to apply knowledge from the human domain to animals as well. The health of the gastrointestinal tract holds the same significance in both humans and animals; thus, we, along with some other authors, focus on the inflammatory diseases of the intestines of animals, broilers specifically [5,6,7,8]. The current study focuses on the health issues of the gastrointestinal tract which are affected by the physical structure of feed in poultry.

The feed particle size influences the feed intake [9,10], growth changes [1], or development of the digestive tract [11]. Several studies on broiler chickens observed that feeding with a coarse mixture increases the relative gizzard weight and decreases pH [12,13,14]. The experiments continued until days 42 [12] and 21 of the broilers’ age [13,14]. A huge, well-developed gizzard enhances the movement of the intestines, promoting intestinal motility [15] and increasing the levels of cholecystokinin [16], which stimulates the secretion of pancreatic enzymes and gastroduodenal reflux [17]. A vital element in the use of nutrients from the feed is the area of the intestinal villi. This is where nutrients are transported into the blood and may be related to the animal’s capacity to intake and absorb nutrients. Higher and wider villi have a greater absorption capacity and consequently, the transport of nutrients on their surfaces. Intestinal villi of broiler chickens morphologically adapt and respond to factors related to the macro- and microstructure of the feed [9,18].

Viscosity is the inherent resistance of a liquid substance. In poultry nutrition, a negative relationship between the representation of the non-starch polysaccharides (NSP), energy, and digestibility of nutrients has been demonstrated, as evidenced by their anti-nutritional character [19]. NSPs form gels in the digestive tract, which significantly increase digestion viscosity [20]. This results in the impairment of nutrient and mineral digestibility (particularly Na, P, Ca), their availability to digestive enzymes, and the mechanical hindrance of nutrient diffusion and movement towards the intestinal wall. As a result, this condition will be negatively reflected in lower weight gain and other disturbances [21].

The goal of the feed industry is to produce homogeneous feed, although the quality of compound feed is affected by various factors, including particle size and shape. Size is usually defined as the average particle size distribution of individual components of feed or as the fineness of feed grinding [9]. Wolf et al. [22] described the specific size as coarse >1.4 mm, medium 0.8–1.4, fine 0.4–0.8, and very fine <0.4 mm. The GMD (geometric mean diameter, also Dgw) and GSD (geometric standard deviation, also Sgw) are calculated to indicate the average particle size (GMD) and the uniformity of particle size (GSD) in the mixture [23]. The optimal particle size distribution should correspond to the physiological needs, which enables the optimal use of nutrients and increases the productivity of the animals. However, recommendations concerning the ideal size of particles vary. In general, finer grinds are reported to increase grinding energy consumption and reduce the grinding equipment’s capacity, increasing dust problems. Particles that are too fine are linked to adverse effects on the gastrointestinal tract’s health and performance [14]. On the other hand, the fine particles may improve digestibility and decrease feed consumption.

The aim of this study was to assess how feed particle size in broiler diets affected the morphology and morphometry of the gastrointestinal tract, the digesta viscosity, and the blood biochemical parameters in terms of health, especially the health of the gastrointestinal tract. Since it was a feeding trial, performance parameters, feed consumption, and feed conversion were also evaluated as additional parameters.

## 2. Materials and Methods

### 2.1. Animals and Experimental Conditions

The animal procedures were reviewed and approved by the Animal Care Committee of Mendel University in Brno and by the Ministry of Education, Youth, and Sports (MSMT-17862/2022-4) of the Czech Republic.

In five replicates, 90 one-day-old male Ross 308 breed broiler chickens were divided randomly into three experimental groups, with each group consisting of 30 chickens. A cage system was used. The experiment was conducted with five replicates, and each cage accommodated six birds. The floor area of each cage is 8500 cm^2^, and the maximum stocking density of each cage is 22.60 kg/m^2^. The lighting program, temperature, and humidity levels were set in accordance with the supplied technological instructions [24]. The broilers were initially fed a starter diet from day one to day ten of age. From the eleventh day to the thirty-fifth day of age, the chicks were fed the experimental grower diet. Throughout the feeding period, chickens had ad libitum access to food, with a daily addition of access to a new dose to feed, allowing them to consume food freely, without any imposed restrictions. The feed intake of each group was recorded on daily basis. Twice a week, the feed residues were used to evaluate their feed particle size (after 24 h of feeding). The body weight was noted weekly.

During the experimental trial, three non-pelleted diets were utilized, with the key distinguishing factor being the particle size of the feed. The first experimental group, referred to as the Coarse group, was provided with a diet containing predominantly coarse particles. The Medium group received a diet with a reduced proportion of coarse particles compared to the Coarse group. Lastly, the Fine group received a diet with a fine particle size throughout. A hammer (3 mm and 6 mm sieve) and roller mills were used to ensure the different feed particle sizes. The ingredients in these diets were utilized in equal quantities. This ensured that the isoenergic and isonitrogenous diets remained consistent across all experimental groups. The compounds and chemical composition of the grower diets used in the experiment are detailed in Table 1.

### 2.2. Sieve Analysis

The physical structure of grower diets was assessed using dry sieving with a Retch AS 200 Control separator. A representative sample from each diet, weighing 100 g, was sieved for a duration of 10 min using a set of sieves with mesh sizes of 3 mm, 2 mm, 1.5 mm, 1 mm, and 0.3 mm. The amplitude was set to 1.8 mm/g. After the sieving process, the retained feed on each sieve was determined by subtracting the weight of the sieve and the retained feed from the initial weight of the sieve. The GMD and GSD were calculated [25]. The feed residues left on the sieves after sieving were evaluated using the same procedure. The particle size distribution is illustrated in Figure 1, while Table 2 presents the values of GMD and GSD for the respective diets used.

The feed particle size of feed mixtures differs significantly, which is proven by the values of GMD.

### 2.3. End of the Experiment and Sample Collection

At the end of the trial, the broilers were weighed and then subjected to slaughter and evisceration. Specific parts of the gastrointestinal tract and the liver were selected for further analysis, and their weights and measurements were recorded. In addition, the chyme was collected, and the viscosity and density were measured. We used the same methods for measuring the gastrointestinal tract and the same histomorphological measuring techniques as those used in our previous article [26].

Fifteen chickens from each group (comprising three of each replicate) were chosen and slaughtered by decapitation. The entire gastrointestinal tracts were removed, and the tracts were divided, according to the methods of Amerah et al. (2007) [14], into the following sections: crop, proventriculus, gizzard, duodenum, jejunum, ileum, ceca, and colon. These sections were emptied and the remaining fat and mesenteries were removed. The removed portions from the small intestine encompassed the region stretching from the junction of the gizzard to the pancreatic and bile ducts (duodenum), the area between the end of the duodenum and Meckel’s diverticulum (jejunum), and the segment between Meckel’s diverticulum and the ileo-ceco-colic junction (ileum). The lengths (or widths) and weight of each segment, after being emptied, were meticulously recorded. To determine the gizzard height, the maximum distance between the proximal (distal limit of the proventriculus) and distal (proximal limit of the duodenum) parts of the gizzard was measured. The gizzard width was measured by determining the maximum distance at right angles to the gizzard height. The gizzard depth was evaluated by measuring the greatest distance between tendineal centers on the two flat sides of the gizzard, while the gizzard muscle height was determined by measuring the maximum height of the primary muscle along its greatest extension. All measurements of the gizzard were performed using a slide caliper. The obtained values were recalculated and expressed relative to the body weight of the chickens, with the measurements standardized per 1 kg [14].

In order to determine viscosity, the fresh digesta (from two birds per replicate) was removed from the distal part of the ileum (in the distal part of the ileum, the digestion processes in the small intestine are completed) according to the methods of Yasar (1990) [27]. The digesta was collected in tubes and subsequently centrifuged for 10 min at 3000 rpm. The resulting supernatant was carefully transferred into Eppendorf tubes. The samples were subjected to a dynamic viscosity test using an RST rheometer (Brookfield, MA, USA). The measurement was conducted at a constant shear strain rate of 50 s^−1^ using a standard cone-plate geometric arrangement (RCT-50-2; α = 2°), equipped with a temperature duplicator system. The measurement was performed in 10 replicates at 40 °C, and the sample volume was 1.2 mL.

### 2.4. Sample Collection and Histological Examination

After the slaughter and evisceration of ten broilers, the duodenum, jejunum, and ileum were removed from each one. Samples of 1 × 1 × 1 cm size were taken from defined areas (the mid-gut) of the duodenum (the apex of the duodenum), jejunum (midway between the point of entry of the bile ducts and Meckel’s diverticulum), and ileum (10 cm proximal to the ileocecal junction) for histopathology evaluation. Samples were taken immediately after the slaughter of the experimental animals and fixed in neutral buffered formalin (10%). The 5 μm thick hematoxylin-eosin-stained sections were prepared, and then paraffin embedding and histological processing were performed. The histological sections were evaluated using an OLYMPUS BX40 microscope.

The samples were dehydrated in an ascending graded series of ethanol using a TP 300 tissue processor and embedded in paraffin wax. Serial sections were cut into 5 μm lengths using a microtome PFM ROTARY 3003 PFM. Cross-sections of the ileum were stained in a Tissue-Tek Prisma automat with Meyer’s Hematoxylin and Eosin, according to the standard histological protocol [28,29].

Histomorphic measurement methodology microphotography of all sections submitted for morphometry was conducted using a Canon EOS 2000D camera at 100× magnification, and histomorphometric measurements of intestinal villi lengths and crypt depths were performed via the Quick PHOTO CAMERA 3.2 program. Subsequently, the amount of *Coccidia* sp. was determined quantitatively per 10 high-power fields (HPF) at 40× magnification.

For each sample, the villi height (VH) and depth of crypts (CD) were measured. Only samples with apparently intact, full-sized intestinal villi, showing no signs of autolysis or mechanical damage, were included in the measurements. Five lengths of villi and depths of crypts were measured per animal, resulting in a total of 50 measurements (*n* = 50). The measurement techniques employed were based on methods previously described by other researchers [30,31,32]. Additionally, the villus height to crypt depth (VH:CD) ratio was calculated, which compares the villus height to the crypt depth [33]. All measurements were conducted by the same person to ensure consistency. The ratios between villi length and crypts depth was assessed.

Blood was collected from the vena jugularis into heparinized tubes and centrifuged for 10 min at 3000 rpm. After the completion of sample collection, all samples were centrifuged within a maximum of 2 h. The centrifugation process aimed to separate the blood plasma from the other components. The separated blood plasma was then frozen at a temperature of −20 °C until further biochemical examination. The examination of the blood plasma samples involved determining various parameters using standardized biochemical methods. A total of 10 plasma samples were analyzed to obtain the required data. Erba Lachema (Czech Republic) commercial sets were utilized for this purpose. The biochemical analysis was performed using an Ellipse automatic biochemical analyzer (AMS Spa, Italy): enzymes activity AST—aspartate aminotransferase (AST/GOT 500); GGT—gamma-glutamyltransferase (GGT 250); ALT—alanine aminotransferase (ALT/GPT 500); ALP—alkaline phosphatase (ALP AMP 500); and LD—lactate dehydrogenase (LDH-L 100). As other markers of hepatic metabolism, fat, and nitrogen metabolism, the following markers were determined: concentrations of the total bilirubin—Tbili (BIL T JG 350); uric acid (UA—UA 500, no. 10,010,225 Erba Lachema, Czech Republic); Urea (Urea, no. UR 107; Randox, United Kingdom); TP—total protein (TP 500); and albumin (Alb 500). The globulin content (TP minus albumin) and albumin to globulin ratio were also calculated.

In the next step, the levels of α-1 globulins, α-2 globulins, ß-globulins, and γ-globulins were determined using the Interlab G26 electrophoretic analyzer (Interlab S.r.l., Italy) in combination with the SRE607K set. The process involved separating the proteins through agarose gel electrophoresis at an alkaline pH. Once separated, the gel was denatured and stained with Acid Violet. It was then subjected to decolorization and drying. The quantification of the divided zones on the gel was performed using densitometry, which is integrated into the instrument used for analysis.

### 2.5. Statistical Analysis

The collected data were processed by Microsoft Excel (USA) and TIBCO Statistica version 12.0 (USA). The experimental unit was the data set for body weight, weight gain, feed intake, feed conversion ratio, and the other monitored parameters. The Shapiro–Wilk test was used to assess the normality of the data distribution, and it was determined that the dataset followed a normal distribution. One-way analysis of variance (ANOVA) was performed to examine the differences between groups. Sheffé’s test was used to further analyze the differences between groups. A significance level of *p* < 0.05 was considered statistically significant in determining differences.

## 3. Results

### 3.1. Used Diets vs. Feed Residues

Table 3 shows differences in the distribution of feed particle sizes and the values of GMD and GSD between the diets and their residues. All groups showed the same trend, where particles over 1.5 mm are represented in the feed residues in a smaller proportion than in the original diets (*p* < 0.05). For particles 1–1.5 mm, there is the opposite result for the Coarse and Medium groups, when the ratio in the diets is smaller (*p* < 0.05). This is the same for all groups, even for particles with a size of 0.3–1 mm. Particles smaller than 0.3 mm were represented in the feed residues of all three groups in a greater proportion than in the original mixture (*p* < 0.05). There are significant differences between the coarse diet and the coarse residues in regards to GMD values (*p* < 0.05). The same trend is found in the medium diet and the medium residues (*p* < 0.05). On the other hand, the fine diet compared to the fine residues showed no significant difference.

### 3.2. Morphology of the Gastrointestinal Tract and Viscosity

Table 4 shows the influence of different feed particle sizes on the morphometry of the gastrointestinal tract. Some statistically significant differences were found in the gizzard weight and depth values, where the Coarse and Medium groups showed higher values than did the Fine group (*p* < 0.05). There are also difference between the values of colon weight, where the colon weight from the Medium group exhibited a lower weight than that of the other groups (*p* < 0.05).

Table 5 shows the influence of feed particle size on the villus height and crypt depth in individual parts of the small intestine (duodenum, jejunum, ileum). The data show statistically significant differences between the groups, and the Coarse group exhibited higher villi and deeper crypts in the duodenum part than those measured in the other groups (*p* < 0.05). The opposite effect was found in the jejunum values, in which the Coarse group showed lower values than did the other both groups (*p* < 0.05). The Coarse group also exhibited higher villi in the ileum portion. The VH/CD ratio was higher in the Coarse group in the jejunum and ileum areas (*p* < 0.05).

The Figure 2 shows the microscopic sections of broilers´ duodenum. According to the histomorphological description of the duodenum in the Coarse group, no surface epithelial injury was found. The intraepithelial lymphocytes observed in the sample were found to be within the range of up to 10 per high-power field (HPF). The lacteals, which are lymphatic vessels in the small intestine responsible for absorbing dietary fats, were of a normal diameter, accounting for less than 25% of the villous width. Within the villous lamina propria, the area occupied by lymphocytes and plasma cells was up to 26–50% of the area of one HPF. Granulocytes, a type of white blood cell, were present in quantities of up to 2 per HPF. The dilation of the crypts was observed in up to 2% of the crypts. There was a normal amount of mucosal fibrous substance present, with up to 2 fibrocytes separating the crypts. There were no *Coccidia* sp.—0/10 HPFs. For the Medium group, no surface epithelial injury was found. Intraepithelial lymphocytes were found to be within the range of up to 10/HPF. The lacteals were of normal diameter, which is less than 25% of the villous width. Within the villous lamina propria, the area occupied by lymphocytes and plasma cells was up to 25% of the area of one HPF. Granulocytes were present in quantities of up to 2 per HPF. Crypt dilation was up to 2% crypts. There were normal amounts of mucosal fibrose substance, with up to 2 fibrocytes separating the crypts. *Coccidia* sp. was not found (0/10 HPFs). Similarly, in the Fine group, no surface epithelial injury was found. Intraepithelial lymphocytes were found to be within the range of up to 10/HPF. The lacteals were of normal diameter, at less than 25% of the villous width. Lamina propria lymphocytes and plasma cells occupied 26–50%/HPF. Granulocytes were found in a maximum rate of 10/HPF. Crypt dilation was up to 2% crypts. A normal amount of mucosal fibrose substance was found, which is up to 2 fibrocytes separating the crypts. *Coccidia* sp. was 0/10 HPFs.

In the jejunum part of the Coarse group, no surface epithelial injury was found (see Figure 3). Intraepithelial lymphocytes were present in quantities of up to 10/HPF. The lacteals exhibited a normal diameter, accounting for less than 25% of the villous width. Within the villous lamina propria, the lymphocytes and plasma cells occupied up to 25% of the area of one HPF. Granulocytes were observed in quantities of up to 2/HPF. Crypt dilation was maximally 2% crypts. A normal amount of mucosal fibrose substance was found, with up to 2 fibrocytes separating the crypts. *Coccidia* sp. was not foung in the samples (0/10 HPFs). In the Medium group, there was no surface epithelial injury. Intraepithelial lymphocytes were up to 10/HPF. The lacteals had a normal diameter, at less than 25% of the villous width. Within the villous lamina propria, the lymphocytes and plasma cells occupy 26–50% of the area of one HPF. Granulocytes were observed in quantities of up to 2/HPF. Crypt dilation was up to 2% crypts. There was a normal amount of mucosal fibrose substance, which is up to 2 fibrocytes separating the crypts. There was no *Coccidia* sp. (0/10 HPFs). In the Fine group, no surface epithelial injury was found. Intraepithelial lymphocytes were up to 10/HPF. The lacteals were of normal diameter, at less than 25% of the villous width. Within the villous lamina propria, the lymphocytes and plasma cells occupy up to 25% of the area of one HPF. Granulocytes were up to 2/HPF. Crypt dilation was up to 2% crypts. A normal amount of mucosal fibrose substance was found (up to 2 fibrocytes separating the crypts). *Coccidia* sp. was not found (0/10 HPFs).

Finally, as shown in Figure 4, in the ileum area in the Coarse group, surface epithelial erosion of up to 10% was found. The intraepithelial lymphocytes were 20–30/HPF. The lacteals had a normal diameter, at less than 25% of the villous width. Within the villous lamina propria, the lymphocytes and plasma cells occupied 51–75% of the area of one HPF. Granulocytes were up to 2/HPF. Crypt dilation was up to 2% crypts. There was a normal amount of mucosal fibrose substance, with up to 2 fibrocytes separating the crypts. *Coccidia* sp. was not found in the samples (0/10 HPFs). Similarly, in the Medium group, no surface epithelial injury was found. The intraepithelial lymphocytes were up to 10/HPF. The lacteals had a normal diameter of less than 25% of the villous width. Within the villous lamina propria, the lymphocytes and plasma cells occupied up to 26–50% of the area of one HPF. Granulocytes were found up to 2/HPF. Crypt dilation was present in up to 2% crypts. There was a normal amount of mucosal fibrose substance, with up to 2 fibrocytes separating the crypts. *Coccidia* sp. was not found (0/10 HPFs). Furthermore, in the Fine group, no surface epithelial injury was found. Intraepithelial lymphocytes were observed in quantities of up to 10/HPF. Lacteals had a normal diameter of less than 25% of the villous width. Within the villous lamina propria, the lymphocytes and plasma cells occupied up to 25% of the area of one HPF. Granulocytes were up to 2/HPF. Crypt dilation was up to 2% crypts. A normal amount of mucosal fibrose substance was found, with up to 2 fibrocytes separating the crypts. No *Coccidia* sp. was found in the samples (0/10 HPFs).

The viscosity and density values were similar in all groups (Table 6). This indicates that the feed particle size had no effect on the viscosity and the density of the digestion in this experiment.

The selected blood biochemical parameters are shown in Table 7. According to these values, the feed particle size of the diets influenced the level of GGT and urea, whereby the GGT value was increased in the Fine group compared to the values in the other groups (Medium and Coarse), and urea in Fine group, contrastingly, was decreased compared to the values in the coarse group (*p* < 0.05). The differences in the values of the other parameters were not significant.

Table 8 shows the mean values of feed consumption divided into three parts. The first values are for the starter period (first 10 days), the second for the grower period (10–35 days), and the last for the whole duration of the fattening period (1–35 days). The total feed intake per whole experiment was 2985, 2884, and 3026 g/bird (Coarse, Medium, and Fine, respectively). No differences were observed between the groups.

Table 9 shows the broiler body weight during the experiment. No statistically significant differences were found in the average weights in the individual weekly weights.

No statistically significant differences were observed between the groups in regards to carcass traits. This indicates that there was no influence of feed particle size in the diet on these parameters (Table 10).

## 4. Discussion

### 4.1. Diets vs. Feed Residues

To evaluate the feed particle size in the diets and the feed residues, the evaluation on a set of sieves was used, and then the GMD and GSD were calculated as the most used indicators of the feed particle size. Different diets were fed to the broilers, and the residues of the presented feeds were evaluated by the particle size and compared with the size in the original diets. The data of our experiment demonstrate that chickens select larger particles from feed diets. The preference was for particles larger than 1.5 mm. The finest particles (smaller than 0.3 mm) were also preferred. In this case, it could be because such small particles were adhered to the large particles (due to the addition of oil to the diets) and eaten along with them. Part of the finest particles may also have been lost during sampling handling due to the high dustiness of these particles. The GMD values for the Coarse and Medium groups decreased compared to those of the original diets. For the group consuming the fine mash diet, the values were practically identical. The findings of our experiment confirmed the results of some previous authors [9,34] that broilers, or poultry in general, prefer coarser particles to finer ones. Amerah et al. [9] used diets processed by wheat grinding in the hammer mill to pass through 3 mm and 7 mm sieves. This led the separation of some feed components from the diet, and the broilers would not receive the complete mixture. This could subsequently influence other monitored parameters. The GMD values in our experiment for the Coarse, Medium, and Fine group were 1111.26 vs. 959.89 vs. 730.48 µm, which corresponds to the results of Ege et al. [35], who used 2 types of diets (707 vs. 1096 µm).

### 4.2. Morphology of the Gastrointestinal Tract and Viscosity

The particle size can affect the size of different parts of the digestive tract [11,36]. The gastrointestinal tract, particularly the gizzard, quickly adapts and reacts to changes in the composition of the diet. Studies showed a rapid increase in the gizzard weight when some structural components, such as oat hulls, shavings, sugarcane bagasse, and whole or coarse grain particles, are included in the feed [36,37,38]. Adding structural components to the diet can even double the original gizzard size [39]. Nir et al. [34,40] claim that coarse particles have a positive impact on the growth of the gizzard, which they proved in their studies. The weight of the gizzard was increased by 26% and 41%. Our study confirms these findings, as the gizzard weight in the Medium group was 11.5% greater, and in the Coarse group, 12.1% greater, than that in the Fine group. Ege et al. [35] in their experiment on laying hens noticed a 55% increase in the gizzard weight when the GMD was increased. Biggs and Parsons [41] observed a gizzard weight increase in broilers fed a proportion of wheat grains as early as the seventh day of age. Conversely, if the diet was very finely ground, there was a negative effect on the size of the gizzard and the overall development of the digestive tract and intestinal function. In this case, the gizzard remains relatively underdeveloped and, on the contrary, the crop is enlarged [42]. In contrast, some studies carried out on poultry found no changes in the gizzard in connection with the change in the coarseness of the feed particles [43,44]. We also found a greater colon mass in the Medium vs. the Fine group. However, to our knowledge, this trend was not noted by other authors dealing with the same issue. No demonstrable changes were recorded in any other section of the gastrointestinal tract in our experiment, which corresponds with the results of the study by Naderinejad et al. [45] on broiler chickens. They showed that maize particle size in the diet did not influence the relative weight or length of GIT parts, except for the gizzard. In contrast, Amerah et al. [14] found shorter individual intestinal segments in a group fed a coarser mixture than in a group fed a diet with a medium particle size. According to Ege et al. [35] there is an increase in the weight of the liver and a lengthening of the small intestine in laying hens fed with finer crumbly mixtures. In this way, the negative effects of the faster passage of feed through the gizzard are solved, simultaneously, limiting its mechanical functions on digestion and absorption of nutrients [36].

Another investigated parameter which can be influenced by the structure of the feed is the intestinal surface. The length of the villi and the depth of the crypts, specifically, their area, is an important aspect for the utilization of nutrients from the feed. This may be related to the nutrient intake that the animal is able to absorb [46]. In our study, villus and crypt parameters in the small intestine were affected by the feed particle size in the diet, when the Coarse group exhibited higher villi and deeper crypts in the duodenum than did the other groups. The opposite effect was found in the jejunum values, in which the Coarse group showed lower values than did either of the other groups, but the VH/CD ration was higher. The Coarse group also exhibited higher villi and a higher VH/CD ratio in the ileum part. The intestinal villi of broiler chickens have been shown to morphologically adapt and respond to factors related to the macro- and microstructure of the feed [14,18]. To our knowledge, studies reporting the effect of feed particle size on the intestinal morphology are limited. Previous studies usually used pelleted and mashed forms of feed for their comparisons [14,45,47]. The findings of Nadeinejad et al. [45] are more similar to those of Amerah et al. [14], which showed positive effect on the villus height and crypt depth ratio of broilers fed a pelleted diet compared with birds fed the mash diet. Ege et al. [35], in their experiment on laying hens fed a pelleted diet, recorded higher and wider villi, as well as the ratio between the height of the villi and the depth of the crypts.

The viscosity of the chyme is one of the most important parameters in the evaluation of animal nutrition. In our experiment, the digesta viscosity in broilers was not affected by different feed particle sizes in the diet. Yasar [48] mentioned a high value of digesta viscosity in poultry fed with a fine mixture compared to those fed with diets containing medium or coarse wheat particles.

### 4.3. Blood Biochemical Parameters

Elevated levels of GGT (gamma-glutamyltransferase) and the simultaneous reduction in blood urea can be caused by various factors and conditions. GGT is an enzyme primarily present in the liver and kidneys, while urea is a waste product of nitrogen metabolism produced in the liver and excreted by the kidneys. There are several possible causes of elevated GGT levels. Increased GGT levels are often associated with liver damage, such as hepatitis, liver cirrhosis, or other forms of liver disease. GGT is frequently used as a marker of liver damage. Another possibility is choledochal disease. The choledochus is the bile duct that connects the liver to the small intestine. Dysfunction of this duct, such as the presence of gallstones or tumors, can lead to increased GGT levels. Elevated GGT levels could be a marker of oxidative stress and subclinical inflammation [49]. Impaired urea synthesis in the liver can occur in various conditions, such as liver failure, cirrhosis, or hepatitis. In these conditions, the normal liver function can be disrupted, leading to reduced urea synthesis. This can result in lower blood urea levels. Reduced blood urea levels can be caused by impaired kidney function. If the kidneys are unable to efficiently filter the blood and eliminate nitrogenous waste substances, it can result in decreased blood urea levels [50]. It is important to realize that elevated GGT levels and decreased urea levels are not specific indicators of a particular diagnosis but can indicate a wide range of possible conditions or diseases.

### 4.4. Performance Parameters

Feed consumption is often associated with the selection of coarse particles from the diet. Some authors noted increased feed intake with a coarse feed mixture compared to a fine feed [9,51]. We did not find this trend in our experiment, and the differences in the consumption were not significant; it was the same with the feed conversion. Other studies, however, confirm our conclusions, and also do not mention any differences in consumption depending on the feed particle size [52,53].

The different feed particle size did not influence the body weight during the experiment. The same trend was observed for slaughter yield and related parameters, which does not correspond to the findings of other authors [9,51]. Moreover, Hamilton and Proudfoot [54] reported improved weight gain when broilers were fed a mixture containing coarse particle sizes. Broilers fed a coarse diet had a higher body weight at the end of an experiment in the study of Abd El-Wahab et al. [55]. In their experiment, they used two types of diets (coarse and fine), and the percentage of particles greater than 2 mm significantly differed between the fine and coarse diets. For example, Lv et al. [56] did not find any significant effect on broiler performance parameters during the whole experiment. In their experiment, they used different particle sizes for corn (573, 865 vs. 1027 µm), wheat (566, 1110 vs. 1183 µm), and soybean meal (490, 842, vs. 880 µm). In the study on cockerel chickens, Sogunle et al. [57] did not mention any changes in performance parameters such as final body weight, leg, weight, and thigh weight, nor were there any differences in feed intake, FCR, or weight. According to Chewning et al. [58], the smaller particle size (screen of 1.66 mm) of maize in comparison with the coarse particles (screen 7.9 mm), in mash form, decreased the feed per unit gain by 31 and 17 points during 0 to 21 and 0 to 44 days, respectively.

## 5. Conclusions

The current study demonstrated that broilers fed a non-pelleted diet prefer coarse particles. The use of coarse feed particle size in the diet had a positive effect on the gizzard weight and small intestinal villi height and crypt depth, which increased the surface area intended for digesting nutrients. On the other hand, the use of finely ground particles in the feed increased the level of gamma-glutamyl transferase and at the same time, decreased the level of urea, which could indicate adverse changes in the liver. However, it is important to realize that an increased level of GGT and a decreased level of urea are not specific signs of one specific diagnosis. At the same time, body weight gain, as well as the feed intake and feed conversion ratio, as basic markers of performance, were not affected by using different feed particle sizes. In conclusion, it can be stated that feed mixtures with a higher proportion of coarse particles positively affect the development and health of the gastrointestinal tract in broilers because a healthy and well-developed gastrointestinal tract is the basic prerequisite for successful production. Although the study shows positive results in broiler chickens, it would be appropriate to extend the study and monitor the effect of feed particle size in other species as well. It would be also appropriate to increase the number of individuals in the study to obtain more relevant results, since for some examinations (e.g., histomorphological, viscosity), a lower number of animals (10 per group) was used.

## Figures and Tables

**Figure 1 animals-13-02532-f001:**
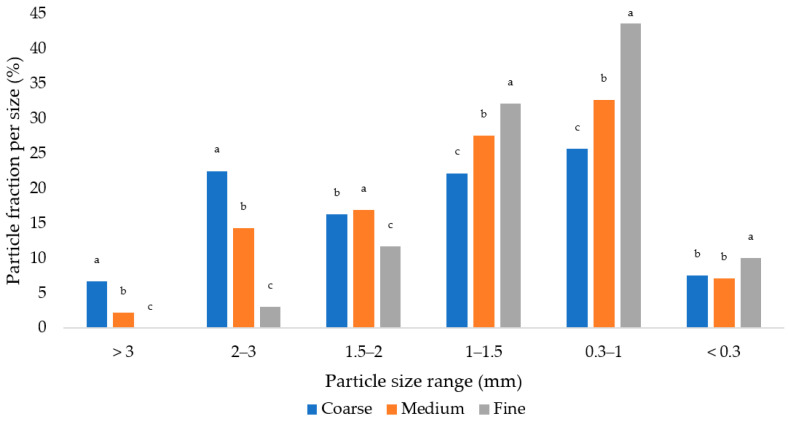
Feed particle size distribution of used diets. ^a,b,c^—different letters indicate statistically significant differences (*p* < 0.05).

**Figure 2 animals-13-02532-f002:**
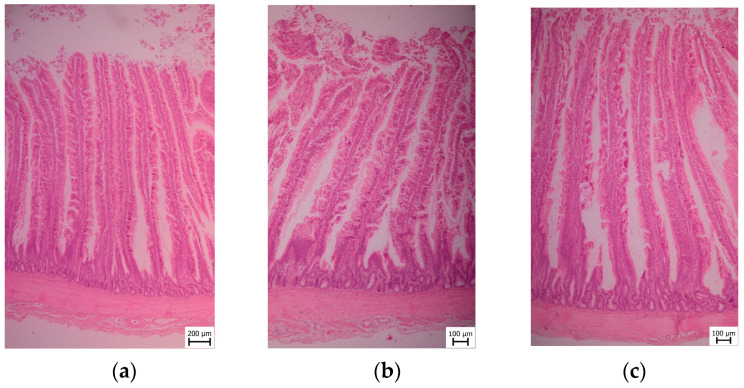
Microscopic section of broilers´ duodenum segments of Coarse, Medium, and Fine groups. (**a**) Coarse; (**b**) Medium; (**c**) Fine.

**Figure 3 animals-13-02532-f003:**
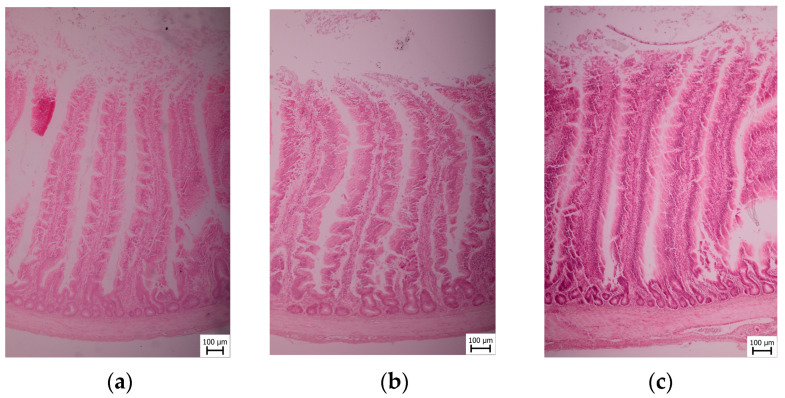
Microscopic section of broiler jejunum segments of Coarse, Medium, and Fine groups. (**a**) Coarse; (**b**) Medium; (**c**) Fine.

**Figure 4 animals-13-02532-f004:**
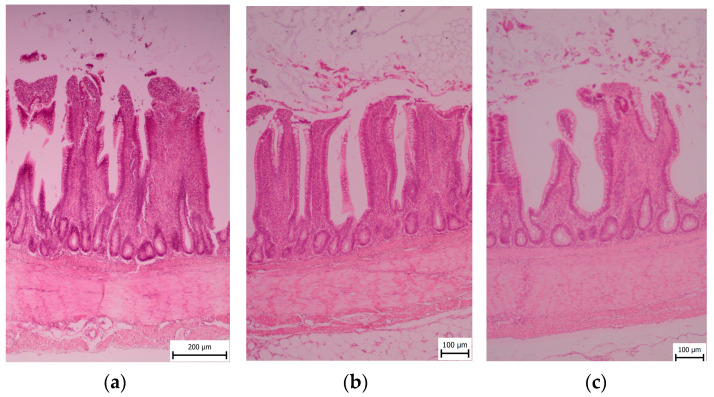
Microscopic section of broiler ileum segments of Coarse, Medium, and Fine groups. (**a**) Coarse; (**b**) Medium; (**c**) Fine.

**Table 1 animals-13-02532-t001:** Components and chemical analysis of used grower diets in 100% dry matter.

Component (g/kg)	Coarse	Medium	Fine
Maize	366.4	366.4	366.4
Soybean meal	396.0	396.0	396.0
Wheat	153.2	153.2	153.2
Rapeseed oil	40.0	40.0	40.0
Premix	30.0	30.0	30.0
Limestone, milled	4.0	4.0	4.0
Monocalcium phosphate	5.9	5.9	5.9
DL-Methionine	1.5	1.5	1.5
Chromium oxide	3.0	3.0	3.0
ME_N_ (MJ/kg) *	12.3	12.3	12.3
Crude protein	244.8	244.4	244.7
Ether extract	83.8	83.1	83.5
Crude fiber	54.1	53.3	48.2
Crude ash	69.6	68.4	68.9

Legend: premix contained (per kg): L-lysine 2.58 g; DL-Methionine 2.52 g; Threonine 1.47 g; calcium 5.04 g; phosphorus 1.65 g; sodium 1.38 g; copper 15 mg; iron 75 mg; zinc 99 mg; manganese 99 mg; iodine 0.9 mg; selenium 0.36 mg; retinol 9900 IU (international units); calciferol 5001 IU; tocopherol 45 mg; phylloquinone 1.5 mg; thiamine 4.2 mg; riboflavin 8.4 mg; pyridoxine 6.0 mg; cobalamin 28.8 µg; biotin 0.18 mg; niacinamide 36 mg; folic acid 1.71 mg; calcium pantothenate 13.35 mg; and choline chloride 180 mg. * Apparent metabolize energy (calculated value).

**Table 2 animals-13-02532-t002:** GMD and GSD values of used diets.

	*n*	GMD (µm)	±	GSD (µm)
Coarse	10	1111.26 ^a^	±	1085.50
Medium	10	959.89 ^b^	±	847.48
Fine	10	730.48 ^c^	±	604.12
SEM		0.03		
*p* value		0.00		

^a,b,c^—different letters within a column indicate statistically significant differences (*p* < 0.05); GMD—geometric mean diameter; GSD—geometric standard deviation; SEM—standard error of the mean.

**Table 3 animals-13-02532-t003:** Feed particle distribution and the GMD and GSD values of used diets and their residues.

	Coarse				Medium				Fine			
Size (mm)	Diet	Residues	SEM	*p*	Diet	Residues	SEM	*p*	Diet	Residues	SEM	*p* Value
*n*	10	30			10	30			10	30		
>3	6.66 ^a^	4.19 ^b^	0.25	0.00	2.15 ^a^	1.36 ^b^	0.08	0.00	0.00	0.00	0.00	-
2–3	22.48 ^a^	16.70 ^b^	0.65	0.00	14.27 ^a^	9.91 ^b^	0.41	0.00	3.02 ^a^	2.02 ^b^	0.09	0.00
1.5–2	16.29 ^a^	14.96 ^b^	0.24	0.02	16.91 ^a^	11.64 ^b^	0.27	0.00	11.64 ^a^	8.66 ^b^	0.29	0.00
1–1.5	22.17 ^b^	24.22 ^a^	0.19	0.00	27.52 ^b^	29.92 ^a^	0.21	0.00	32.16	32.57	0.34	0.61
0.3–1	25.70 ^b^	35.14 ^a^	1.00	0.00	32.65 ^b^	38.40 ^a^	0.68	0.00	43.60 ^b^	48.97 ^a^	0.63	0.00
<0.3	7.51 ^a^	5.15 ^b^	0.22	0.00	7.05 ^a^	5.51 ^b^	0.14	0.00	9.96 ^a^	7.69 ^b^	0.20	0.00
GMD	1.11 ^a^	1.02 ^b^	0.02	0.01	0.96 ^a^	0.91 ^b^	0.01	0.01	0.73	0.72	0.00	0.51
GSD	1.09	0.87			0.85	0.72			0.60	0.53		

^a,b^—different letters within a row indicate statistically significant differences (*p* < 0.05); GMD—geometric mean diameter; GSD—geometric standard deviation; SEM—standard error of the mean.

**Table 4 animals-13-02532-t004:** Individual sections of the gastrointestinal tract of broilers.

	Coarse	Medium	Fine		
*n*	15	15	15	SEM	*p* Value
	Mean				
Crop (g/kg of BW)	3.05	2.93	3.39	0.11	0.21
Proventriculus (g/kg of BW)	3.73	3.59	3.31	0.08	0.34
Gizzard (g/kg of BW)	14.97 ^a^	14.56 ^a^	11.84 ^b^	0.39	0.00
Gizzard height (mm/kg of BW)	26.40 ^a^	26.87 ^a^	24.19 ^b^	0.47	0.04
Gizzard width (mm/kg of BW)	18.95	19.32	16.87	0.56	0.15
Gizzard depth (mm/kg of BW)	13.93	14.02	12.33	0.32	0.55
Gizzard muscle height (mm/kg of BW)	6.40	6.10	5.73	0.18	0.33
Duodenum (g/kg of BW)	5.88	6.12	6.26	0.16	0.62
Duodenum (mm/kg of BW)	141.77	152.25	139.09	3.59	0.29
Jejunum (g/kg of BW)	11.36	10.94	10.41	0.27	0.37
Jejunum (mm/kg of BW)	330.33	344.84	320.15	6.89	0.35
Ileum (g/kg of BW)	7.75	8.09	7.82	0.20	0.77
Ileum (mm/kg of BW)	343.15	353.45	342.70	7.22	0.80
Colon (g/kg of BW)	1.27 ^b^	1.20 ^b^	1.46 ^a^	0.04	0.04
Colon (mm/kg of BW)	42.43	39.77	37.01	1.30	0.24
Ceca (g/kg of BW)	3.82	3.45	3.62	0.08	0.15
Ceca (mm/kg of BW)	81.62	80.33	79.03	1.55	0.80
Liver (g/kg of BW)	22.32	22.28	23.11	0.25	0.93

^a,b^—different letters within a row indicate statistically significant differences (*p* < 0.05); SEM—standard error of the mean; BW—body weight.

**Table 5 animals-13-02532-t005:** The average duodenum, jejunum, and ileum villus height and crypt depth.

	Coarse	Medium	Fine		
*n*	50	50	50	SEM	*p* Value
	Mean (µm)				
Duo villi	1851.90 ^a^	1796.90 ^b^	1731.36 ^b^	18.06	0.02
Duo crypts	129.40 ^a^	125.18 ^b^	116.16 ^b^	1.80	0.01
Duo VH/CD	14.67	14.81	15.34	0.26	0.53
Jej villi	1043.46 ^b^	1049.08 ^a^	1162.74 ^a^	20.01	0.02
Jej crypts	86.52 ^c^	103.84 ^b^	115.54 ^a^	2.10	0.00
Jej VH/CD	12.34 ^a^	10.53 ^b^	10.14 ^b^	0.20	0.00
Ile villi	494.10 ^a^	394.86 ^b^	410.80 ^b^	8.82	0.00
Ile crypts	80.78	79.92	84.78	1.05	0.14
Ile VH/CD	6.13 ^a^	5.01 ^b^	4.94 ^b^	0.11	0.00

^a,b,c^—different letters within a row indicate statistically significant differences (*p* < 0.05); SEM—standard error of the mean; duo—duodenum; jej—jejunum; ile—ileum; VH—villi height; CD—crypt depth.

**Table 6 animals-13-02532-t006:** Viscosity and density of digestion.

		Viscosity (mPa·s)	Density (kg·m^−3^)
	*n*	Mean	
Coarse	10	5.21	1032.93
Medium	10	4.91	1030.52
Fine	10	5.09	1031.19
SEM		0.13	0.63
*p* value		0.44	0.28

SEM—standard error of the mean.

**Table 7 animals-13-02532-t007:** Blood biochemical parameters.

	Coarse	Medium	Fine		
*n*	10	10	10	SEM	*p* Value
	Mean				
ALT (µkat/L)	0.08	0.09	0.09	0.01	0.65
AST (µkat/L)	3.79	3.81	3.62	0.09	0.66
GGT (µkat/L)	0.22 ^b^	0.22 ^b^	0.26 ^a^	0.01	0.03
ALP (µkat/L)	192.08	206.27	199.53	12.26	0.90
LD (µkat/L)	63.77	57.67	68.34	2.57	0.24
TB (µmol/L)	4.76	4.41	4.86	0.22	0.69
Urea (mmol/L)	2.06 ^a^	1.84 ^ab^	1.70 ^b^	0.06	0.04
UA ((µmol/L)	345.64	298.26	291.11	13.15	0.19
TP (g/L)	31.89	29.27	31.62	0.56	0.11
Alb (g/L)	17.71	17.00	17.92	0.27	0.36
A/G	1.33	1.42	1.32	0.04	0.52
Alb (%)	0.56	0.58	0.57	0.01	0.60
α-1 glob (%)	0.26	0.25	0.25	0.00	0.64
α-2 glob (%)	0.05	0.05	0.04	0.00	0.71
β-glob (%)	0.09	0.07	0.08	0.01	0.40
γ-glob (%)	0.05	0.05	0.06	0.00	0.18

^a,b^—different letters within a row indicate statistically significant differences (*p* < 0.05); SEM—standard error of the mean; Alb—albumins; ALT—alanine aminotransferase; ALP—alkaline phosphatase; AST—aspartate aminotransferase; A/G—albumins/globulins; GGT—gamma-glutamyltransferase; Glob—globulins; LD—lactate dehydrogenase; TB—total bilirubin; UA—uric acid; TP—total protein.

**Table 8 animals-13-02532-t008:** The effect of different feed particle sizes on feed intake and feed conversion ratios.

		Coarse	Medium	Fine		
		Mean			SEM	*p* Value
Feed intake	Starter (g/day/bird)	22.37	23.18	22.67	0.33	0.63
	Grower (g/day/bird)	110.46	106.08	111.98	2.66	0.68
	Overall (g/day/bird)	85.29	82.40	86.47	1.95	0.71
FCR		1.36	1.34	1.34	0.02	0.27

SEM—standard error of the mean.

**Table 9 animals-13-02532-t009:** The effect of different feed particle sizes on body weight (g).

	Day	1	10	14	21	28	35
	*n*	Mean					
Coarse	29	41	196	315	732	1371	2203
Medium	27	41	195	310	664	1335	2196
Fine	29	40	198	348	766	1427	2326
SEM		0.32	3.12	8.90	19.02	29.57	44.17
*p* value		0.34	0.95	0.19	0.09	0.45	0.40

SEM—standard error of the mean.

**Table 10 animals-13-02532-t010:** The average carcass weight, yield, and the yield of the breast and thighs.

	Coarse	Medium	Fine		
*n*	10	10	10	SEM	*p* Value
	Mean				
Body weight (g)	2349	2335	2386	63.31	0.63
Carcass weight (g)	1562	1550	1703	49.38	0.39
Carcass yield (%)	66.55	66.78	68.70	1.08	0.69
Breast (%)	32.23	30.30	31.20	0.61	0.45
Thighs (%)	24.21	22.67	22.48	0.57	0.41

SEM—standard error of the mean.

## Data Availability

The data presented in this study are available on request from the corresponding author.

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
