# Peer review of "Effect of Different Feed Particle Size on Gastrointestinal Tract Morphology, Ileal Digesta Viscosity, and Blood Biochemical Parameters as Markers of Health Status in Broiler Chickens"

_animals, 2023, doi:10.3390/ani13152532_

Round 1

Reviewer 1 Report (Previous Reviewer 3)

Authors have done corrections.

Author Response

Response to reviewer 1

Open Review

Quality of English Language

( ) I am not qualified to assess the quality of English in this paper
( ) English very difficult to understand/incomprehensible
( ) Extensive editing of English language required
( ) Moderate editing of English language required
( ) Minor editing of English language required
(x) English language fine. No issues detected

Yes

Can be improved

Must be improved

Not applicable

Does the introduction provide sufficient background and include all relevant references?

(x)

( )

( )

( )

Are all the cited references relevant to the research?

(x)

( )

( )

( )

Is the research design appropriate?

(x)

( )

( )

( )

Are the methods adequately described?

(x)

( )

( )

( )

Are the results clearly presented?

(x)

( )

( )

( )

Are the conclusions supported by the results?

(x)

( )

( )

( )

Comments and Suggestions for Authors

Authors have done corrections.

We would like to thank the reviewer for the positive evaluation of the manuscript. We really appreciate his work and the time he spent on the manuscript.

Reviewer 2 Report (Previous Reviewer 2)

Dear Authors,

Line 106-107: Stocking density should be given as live weight per square meter.

Line 324: If the differences between the groups are not significant, it would be more appropriate not to give "P>0.05", throughout the manuscript.

Line 332: If the differences between the groups are not significant, it would be more appropriate not to give "P>0.05".

Line 346: delete “No statistically significant differences (P > 0.05);”, throughout the tables

Line 348: If the differences between the groups are not significant, it would be more appropriate not to give "P>0.05".

Table 9: replace “P” with “P value”, throughout the manuscript.

Table 10: Replace “live weight” with “body weight”.

Author Response

Response to reviewer 2

Open Review

Quality of English Language

( ) I am not qualified to assess the quality of English in this paper
( ) English very difficult to understand/incomprehensible
( ) Extensive editing of English language required
( ) Moderate editing of English language required
( ) Minor editing of English language required
(x) English language fine. No issues detected

Yes

Can be improved

Must be improved

Not applicable

Does the introduction provide sufficient background and include all relevant references?

( )

(x)

( )

( )

Are all the cited references relevant to the research?

( )

(x)

( )

( )

Is the research design appropriate?

( )

(x)

( )

( )

Are the methods adequately described?

( )

(x)

( )

( )

Are the results clearly presented?

( )

(x)

( )

( )

Are the conclusions supported by the results?

( )

(x)

( )

( )

Comments and Suggestions for Authors

Dear Authors,

Line 106-107: Stocking density should be given as live weight per square meter.

            Added.

Line 324: If the differences between the groups are not significant, it would be more appropriate not to give "P>0.05", throughout the manuscript.

Line 332: If the differences between the groups are not significant, it would be more appropriate not to give "P>0.05".

Line 346: delete “No statistically significant differences (P > 0.05);”, throughout the tables

Line 348: If the differences between the groups are not significant, it would be more appropriate not to give "P>0.05".

           Lenes 324, 332, 346, 348 corrected.

Table 9: replace “P” with “P value”, throughout the manuscript.

            Replaced.

Table 10: Replace “live weight” with “body weight”.

            Replaced.

We sincerely thank the reviewer for his comments. All mentioned shortcomings have been resolved and fixed. Thank you.

Reviewer 3 Report (New Reviewer)

The authors presented a paper showing that feed size influences health status of the broiler chicken. In general, the study is interesting and shows some novelty. However, the article can be improved by addressing the following comments and suggestions.

Line 1, 43, etc. – the authors should change “digestive tract” into “gastrointestinal tract”

Line 47 – please provide confirmation (reference) that IBD, IBS or ILD are present in broiler chicken.

Line 95 – the authors mentioned about “main goal”. What are “secondary goals” then?

Line 110 – ad libitum should be written in italics.

Line 153 – Digestive tract is a tract which passes food from mouth to the anus. I have the impression that the authors limit the digestive tract to segments located distally from the stomach.

Line 169 - It is not clear why the authors decided to study the digesta of ileum (but not from the duodenum or jejunum). It must be carefully explained.

Line 178, 281 - From histological point of view there are only four kinds of tissues: epithelial, connective, muscular and nervous. Therefore, such terms as “ileum tissue” , “mucosal fibrose tissue” are not justified.

Line 183 – please change to “micrometers”

Line 203 – which vessel was used to collect the blood?

Table 4 – “cecum” or “ceca”?

Figure 2, 3 and 4 – micrographs are too small to make any judgement. Scale bars are missing. Why these images are called histopathological?

Line 303 – check the language correctness of this sentence.

Table 10 – what “live weight” means?

Line 473 – in the present form the conclusions is just repetition of the results. Please mention about the limitations of the study.

The authors should improve the quality of English Language. As mentioned in the report some sentences are barely understood.

Author Response

Response to reviewer 3

Open Review

Quality of English Language

( ) I am not qualified to assess the quality of English in this paper
( ) English very difficult to understand/incomprehensible
( ) Extensive editing of English language required
(x) Moderate editing of English language required
( ) Minor editing of English language required
( ) English language fine. No issues detected

Yes

Can be improved

Must be improved

Not applicable

Does the introduction provide sufficient background and include all relevant references?

( )

(x)

( )

( )

Are all the cited references relevant to the research?

( )

(x)

( )

( )

Is the research design appropriate?

( )

(x)

( )

( )

Are the methods adequately described?

( )

(x)

( )

( )

Are the results clearly presented?

( )

(x)

( )

( )

Are the conclusions supported by the results?

( )

(x)

( )

( )

Comments and Suggestions for Authors

The authors presented a paper showing that feed size influences health status of the broiler chicken. In general, the study is interesting and shows some novelty. However, the article can be improved by addressing the following comments and suggestions.

Line 1, 43, etc. – the authors should change “digestive tract” into “gastrointestinal tract”

            Changed.

Line 47 – please provide confirmation (reference) that IBD, IBS or ILD are present in broiler chicken.

            Added.

Line 95 – the authors mentioned about “main goal”. What are “secondary goals” then?

            We have edited the sentence so that it is not misleading and does not imply other secondary goals.

Line 110 – ad libitum should be written in italics.

            Corrected.

Line 153 – Digestive tract is a tract which passes food from mouth to the anus. I have the impression that the authors limit the digestive tract to segments located distally from the stomach.

            Thanks for your notice. It has been edited throughout the manuscript.

Line 169 - It is not clear why the authors decided to study the digesta of ileum (but not from the duodenum or jejunum). It must be carefully explained.

            We followed Yasar's (1990) methodology with the fact that the ileum is the distal segment of the small intestine, where the final process of digestion in the small intestine takes place. It was added in MaM section.

Line 178, 281 - From histological point of view there are only four kinds of tissues: epithelial, connective, muscular and nervous. Therefore, such terms as “ileum tissue” , “mucosal fibrose tissue” are not justified.

            The text in this section has been supplied directly from an expert in the field of histology and morphology. We did not interfere with these terms.

Line 183 – please change to “micrometers”

            Changed.

Line 203 – which vessel was used to collect the blood?

V. jugularis was used. Addetd in the manuscrip.

Table 4 – “cecum” or “ceca”?

            Corrected to “ceca”.

Figure 2, 3 and 4 – micrographs are too small to make any judgement. Scale bars are missing. Why these images are called histopathological?

            Micrographs are illustrative only. They were added at the request of one of the opponents. All information is described in the text in the section Morphology of the gastrointestinal tract and viscosity.

            The names of Figures were changed to Microscopic section ...

Line 303 – check the language correctness of this sentence.

            The text in whole section of morphology was modified.

Table 10 – what “live weight” means?

            Corrected to “body weight”.

Line 473 – in the present form the conclusions is just repetition of the results. Please mention about the limitations of the study.

            Sentence about the limits of the study was added. It concerns about the type of used animals. It would be appropriate to continue and extend the study to other animal species as well.

Comments on the Quality of English Language

The authors should improve the quality of English Language. As mentioned in the report some sentences are barely understood. 

Some sentences have been changed for better understanding.

We would like to express our deep gratitude to the reviewer for his valuable comments. We are very pleased that the manuscript seems interesting. We have taken your comments and suggestions to heart and tried to incorporate them into our manuscript. Thank you very much for your suggestions to help us create a better manuscript.

Round 2

Reviewer 3 Report (New Reviewer)

The authors only partially responded to my concerns.

Still, some issues need to be improved.

1. More informative micrographs of histological staining and of higher magnification are lacking.

2. The term "digestive tract" is still present throughout the manuscript.

3.  Confirmation (reference) that IBD, IBS or ILD are present in broiler chicken is still lacking.

4. The authors did not mention about any limitations of the study (for example relatively low number of animals use for histomorphometrical studies -only 6 broilers were dissected out).

Without these correction I can not recommend manuscript for publication.

Author Response

Responses to Reviewer

We would like to sincerely and wholeheartedly express our gratitude to the reviewer of our manuscript for their insightful comments and suggestions.

Yes

Can be improved

Must be improved

Not applicable

Does the introduction provide sufficient background and include all relevant references?

( )

( )

(x)

( )

Are all the cited references relevant to the research?

(x)

( )

( )

( )

Is the research design appropriate?

( )

(x)

( )

( )

Are the methods adequately described?

( )

(x)

( )

( )

Are the results clearly presented?

( )

( )

(x)

( )

Are the conclusions supported by the results?

( )

( )

(x)

( )

Comments and Suggestions for Authors

The authors only partially responded to my concerns.

Still, some issues need to be improved.

  1. More informative micrographs of histological staining and of higher magnification are lacking.

With reference to your requirements, we contacted the histologists who performed the measurements. New figures with higher magnification and supplementary scale bar have been inserted into the manuscript.

  1. The term "digestive tract" is still present throughout the manuscript.

            The term digestive tract has been rewritten to gastrointestinal tract in all sections concerning to our study. However, in some parts of the introduction and discussion, the term digestive is unchanged, due to the preservation of the terminology of the original cited studies.

  1. Confirmation (reference) that IBD, IBS or ILD are present in broiler chicken is still lacking.

The terms IBD, IBS, ILD are terms used in human medicine. In poultry, the issue of intestinal health is studied only at the level of inflammation (authors dedicated to this are cited in the manuscript). And that is why, also with regard to and reference to knowledge from the human environment, we are dedicated to the issue of health at the level of the intestines. We apologize that this was not obvious from the text. We edited the manuscript to make it more obvious.

  1. The authors did not mention about any limitations of the study (for example relatively low number of animals use for histomorphometrical studies -only 6 broilers were dissected out).

            In the Material and Methods section (Sample collection and histological examination), we noted an error in the number of chickens used for histomorphometrical studies. We misquoted the number as 6, but 10 chickens from each group were used, it was corrected in the manuscript. We realize that even 10 pieces is a relatively low number and we have added this study limit to the conclusion.

Thank you again for your feedback. We have tried our best to edit all your comments throughout the manuscript. Thank you for the time you spent on the manuscript.

This manuscript is a resubmission of an earlier submission. The following is a list of the peer review reports and author responses from that submission.

Round 1

Reviewer 1 Report

The publication "Effect of different feed particle size on performance, digestive tract morphology and ileal digesta viscosity in broiler chickens" presents the effect of feed particle size on performance and the digestive tract of chickens. In my opinion, the publication has no practical application. Since the composition of the feed in terms of calorific value is the same, it is unreasonable to study the effect of feed particles on performance. For such research to make sense, it would be necessary to obtain a homogeneous feed, in which most particles would be of the same size, while the authors show in Fig. 1 that, for example, coarse feed contains approx. 20% of particles 2-3 and approx. 25% 0.3-1um. The particle size distribution in the form of a Gaussian curve should be given for each feed. Moreover, what is the point of testing the particle size before feeding the birds and then in the leftover feed? Chickens can scatter the feed, and smaller particles can cause dust.

The authors report the influence of particle size on the length and weight of individual sections of the gastrointestinal tract. What is the purpose of this? How can the size of feed particles affect the length of individual sections of the digestive tract? How to accurately separate the individual parts when there is no sharp line between jejunum and illeum etc.?

The authors write about histomorphological studies, but there are no photos of these studies in the paper.

Reviewer 2 Report

Dear Authors,

Please explain how your research differs from earlier research on particle size. Indicate in detail which shortcomings in this regard can be solved.

Why were animals raised for 35 days? It has been found that the target body weight has not been reached, and the feed efficiency utilization has been found to be very low in terms of sector.

Line 25: Replace “live weight” with “body weight”.

Line 85-86: Why did you use so few animals in the experimental groups? Six animals in subgroups are very few. The death of one animal means that the mortality rate is about 15%.

Line 87: Are the broilers reared in cages? Please specify.

Line 87: Stocking density and cage characteristics should be given.

Line 92: “The body weight was regularly 92 noticed.”. Daily or weekly?

Line 103, Table 1: Why were the protein values different even though the diet contained the same ingredients and ratios?

Line 118: Replace “figure 1” with “Figure 1”, and “table 2” to “Table 2”

Line122: Table 2: Delete “c” for “604.12c

Line 148: Replace “live weight” with “body weight”.

Table 3: Some upper lettering has not been made in Table 3.

Table 4: Why didn't you calculate the organ weights in terms of body weight (%)? In the material and method section, it was written that it was determined according to organ weights. Organ weights should be given as %.

Tables: In the lettering of the groups, it is more appropriate to letter the group with the highest number "a".

Table 5: Replace “VL” with “VH”, and throughout the whole manuscript.          

Lines 211-212: It would be more appropriate to write the P values at the end of the sentence.

Lines 214-125: It would be more appropriate to write the P values at the end of the sentence.

Lines 276-277: It would be more appropriate to write the P values at the end of the sentence.

Line 277: Delete “see”

Reviewer 3 Report

There are some comments about this manuscript that have shown below:

Line 49: Authors have to explain more about that references, for example broilers or layer!! and in which age?

Line 103: Authors have to describe starter and grower feed, so if this is starter still have high protein ratio!

Line 151: Authors should cite properly! they have to mention name of Authors and the year of publication

Line 204: significant letter should be super script.

Line 204: The big number should have letter a and the small number has b for superscript 

Line 205: Authors must mention different letters within row or column!

Line 213: It should be better if authors interpretation the result then put table after that interpretation for new table. 

Line 221: Again authors used letter (b) to the highest value!!!

Line 222: Authors must mention the different letters within row or column!!!

Line 284: The title of this table is not clear. Generally this table is not clear authors should re-arrange again so as to feed consumption and FCR be clear!! one more thing, what does mean Trial here!

Line 288: I should be better if change the title to (The effect of different feed particle size on body weight).
